# Wheat Drought Tolerance: Unveiling a Synergistic Future with Conventional and Molecular Breeding Strategies

**DOI:** 10.3390/plants14071053

**Published:** 2025-03-28

**Authors:** Charan Singh, Sapna Yadav, Vikrant Khare, Vikas Gupta, Madhu Patial, Satish Kumar, Chandra Nath Mishra, Bhudeva Singh Tyagi, Arun Gupta, Amit Kumar Sharma, Om Prakash Ahlawat, Gyanendra Singh, Ratan Tiwari

**Affiliations:** 1ICAR-Indian Institute of Wheat and Barley Research, Karnal 132001, Haryana, India; sapnayadav173@gmail.com (S.Y.);; 2Nuclear Agriculture and Biotechnology Division, Bhabha Atomic Research Centre, Mumbai 400085, Maharashtra, India; 3ICAR-Indian Institute of Agricultural Research-Regional Station, Shimla 171001, Himachal Pradesh, India

**Keywords:** drought, wheat, breeding approaches, genomic selection

## Abstract

The development of wheat cultivars capable of withstanding drought conditions is necessary for global food security. Conventional breeding, emphasizing the exploitation of inherent genetic diversity by selecting wheat genotypes exhibiting superior drought-related traits, including root architecture, water use efficiency, and stress-responsive genes, has been used by breeders. Simultaneously, molecular techniques such as marker-assisted selection and gene editing are deployed to accelerate the identification and integration of specific drought-responsive genes into elite wheat lines. Cutting-edge genomic tools play a pivotal role in decoding the genetic basis of wheat drought tolerance, enabling the precise identification of key genomic regions and facilitating breeding decisions. Gene-editing technologies, deployed judiciously, ensure the targeted enhancement of desirable traits without compromising the overall genomic integrity of wheat varieties. This review introduces a strategic amalgamation of conventional and molecular breeding approaches for developing drought-tolerant wheat. The review aims to accelerate progress by seamlessly merging traditional breeding methods with advanced molecular tools, and it also underscores the potential of a synergistic future for enhancing wheat drought resilience, providing a roadmap for the development of resilient wheat varieties essential for sustainable agriculture in the 21st century.

## 1. Introduction

Due to its use as a source of calories and protein, bread wheat (*Triticum aestivum* L.) is an important agricultural crop that contributes to global economies and is essential to maintaining food security [1]. Common wheat, or bread wheat (*Triticum aestivum* L., 2n = 6x = 42, AABBDD), is an allohexaploid species that is primarily autogamous and a member of the Triticeae tribe of the Poaceae family of grasses [2,3]. In many regions of the world, its nutrient-rich grain plays an essential part in daily caloric intake and meets the dietary demands of 2.5 billion people [4]. Globally, the prime challenge is the insufficient availability of water for crop cultivation. In the context of global climate change, there is a continual rise in temperature and depletion of water resources, thereby constraining agricultural productivity [5]. Large regions of Europe, Africa, Asia, Oceania, South America, Central America, and North America are all affected by drought stress (DS), which is one of the main and pervasive threats to cereal productivity, resulting in losses in grain yield (GY) [6,7,8]. The worldwide issue of water scarcity poses a significant concern for sustainable agricultural production, as highlighted by [9]. In light of projections from CIMMYT, a necessary 70% surge in wheat production by 2050 is predicted to ensure food security for a growing global population [10,11]. Therefore, breeding programs specifically targeting the development of wheat cultivars with enhanced drought tolerance are imperative [12]. In 2050, the projected global wheat demand compels wheat researchers to utilize advanced tools for deciphering the complex wheat genome [13]. This accelerates the creation of high-yielding varieties adaptable to challenging environments.

Drought, or moisture stress, significantly affects wheat during crucial stages such as jointing, flowering, and milking. The scarcity of water, essential for physiological processes, imposes constraints on photosynthesis, respiration, dry matter content, and the availability of plant nutrients. Drought-induced effects include stomatal closure, reduced water content, and loss of turgor, sometimes resulting in plant mortality due to the disruption of physiological processes [14]. Disputes remain over which of these traits are most desirable. In light of these controversies, this review proposes the most suitable selection criteria based on the available literature by identifying the best trait combinations [15].

## 2. Wheat Growth Stages and Response to Drought

Drought stress impacts various growth stages, including germination, seedling formation, root and shoot growth, tillering, flowering, pollination, fertilization, seed yield, and quality [16]. Susceptibility to drought exists throughout all the growth periods [17], with distinct effects on vegetative (germination, seedling, tillering) and generative phases (flowering, fertilization, seed formation, grain-filling) [18,19]. The selection of desirable traits under drought at different stages is summarized in Table 1 and further discussed below:

### 2.1. Drought at Seedling Stage

Proper crop establishment, which is the basis for the expression of yield potential, is seriously threatened by early drought. Under situations of water shortage, important parameters that define early vigor, such as root length, coleoptile length, germination percentage, seedling vigor index, and seed water absorption, might be inhibited [20,21,22]. Proper crop establishment, which is the basis for the expression of yield potential, is seriously threatened by early drought. Under situations of water shortage, important parameters that define early vigor, such as root length, coleoptile length, germination percentage, seedling vigor index, and seed water absorption, might be inhibited [20,21,22].

### 2.2. Drought at Tillering Stage

Early-season drought reduces the number of productive tillers, affecting spike development and, ultimately, yield [20,23,24]. While selecting for high tiller numbers is advantageous in early drought, it may be disadvantageous under terminal drought, leading to increased water consumption and limiting the stored soil water during terminal drought [25,26]. Fewer tillers characterize semi-dwarf high-yield genotypes, optimizing resource allocation to structural carbohydrates and maximizing the Harvest Index (HI). Late heading and flowering, coupled with a short grain-filling period, correlate with higher yield during early-season drought [27]. Tailoring trait combinations for specific drought scenarios is crucial, as a ‘xerophytic’ breeding strategy targeting evapotranspiration under extreme harsh environments. However, the variable timing and intensity of stress events in drought-prone areas necessitate ‘adaptable’ breeding strategies [28].

### 2.3. Drought at Grain Filling Stage

In terminal drought, prevalent in the Mediterranean climate, semi-dwarf wheats with dwarfing alleles, early heading, and early maturity traits show an advantage, expressing high yield potential without prolonged stress [29,30,31]. The relationship between primary leaf (PL) and yield under drought stress (DS) conditions varies across environments. In instances of water shortage, a significant correlation between PL and GY is not observed [31,32,33]. Furthermore, a notable reduction in PL attributed to DS is not consistently evident, as the peduncle attains its maximum duration immediately following anthesis [33]. The other trait, leaf rolling (LR) poorly correlates with leaf water potential and water loss rate, suggesting delayed onset due to effective osmotic adjustment [34,35].

## 3. Adaptations for Drought Stress

The drought tolerance mechanisms in wheat are depicted in Figure 1, illustrating the key adaptive strategies involved. Wheat, like many crops, exhibits diverse adaptations to combat drought stress, impacting growth, yield, and quality negatively. Some varieties escape drought through accelerated life cycles, aided by increased leaf nitrogen, higher photosynthesis, and efficient remobilization of photosynthates. Other cultivars are bred for drought tolerance, emphasizing water use efficiency and stress-responsive genes. Strategies encompass deep rooting, enhanced root length and density, reduced transpiration through stomatal closure, leaf rolling, and cuticle wax formation, aligning lifecycle modifications with rainfall patterns have been highlighted by different authors [36]. Dehydration tolerance involves partial dehydration, with mechanisms such as improved germination, stem reserves, prolonged flag leaf duration, osmotic adjustment, and membrane integrity maintenance.

### 3.1. Morpho-Physiological Adaptation

The drought stress induces morphological changes in wheat (Table 2), notably reducing plant height (PH) and peduncle length (PL) due to protoplasm dehydration, leading to turgor loss, and impaired cell expansion and cell division [37,38,39]. Peduncle length (PL) is often shortened by water deficiency and the last internode of the main stem beneath the spike is also crucial for supporting grain filling (GF) through assimilate remobilization, particularly in DS conditions [33,40]. Leaf rolling (LR) occurs due to turgor loss and inadequate osmotic regulation in the tissues of leaves [41]. LR reducing solar radiation exposure by 41–48%, lowers leaf temperature and transpiration, creating a humid microclimate that aids photosynthesis and atmospheric water interception [34,42,43]. While LR is recognized as a drought avoidance mechanism, its impact on wheat yield has not been extensively investigated, and results are not consistently conclusive [34,43,44]. PH, PL, and LR serve as widely adopted traits for large-scale wheat phenotyping, thereby drawing attention of breeders for exploring their potential as reliable screening indicators.

Reducing lateral root branching density in topsoil and increasing root hairs can enhance subsoil water uptake, minimizing root metabolism. Excessive roots in dry topsoil may elevate abscisic acid levels, potentially reducing stomatal conductance and crop yield [45]. To improve drought resistance, selecting for reduced root angle, axial root number, and branch density can promote effective root growth. Deep-rooted cultivar selection criteria encompass root angle, axial root number, and branching, emphasizing the importance of steep root growth angles [46]. The wheat plants respond to moisture deficit through physiological adaptations also, including stomatal closure to reduce water loss, decreased growth rate and leaf expansion, and adjustments in respiration and energy consumption for resource conservation. Stomata, pores on leaf surfaces, facilitate CO_2_ entry for photosynthesis but also contribute to water loss through transpiration. The challenge lies in balancing the need for CO_2_ in photosynthesis with preventing leaf tissue drying. Breeding programs leading to the development of wheat cultivars with delayed leaf senescence, extending photosynthesis duration, and improving water use efficiency (WUE) will be a boon for enhanced wheat biomass and grain production under drought stress conditions.

### 3.2. Cellular and Biochemical Adaptation

Wheat’s cellular and molecular mechanisms for drought resistance are illustrated in Figure 2. Drought stress adversely impacts plant growth by affecting physiological and biochemical processes, including respiration, translocation, ion uptake, water potential, stomatal closure, photosynthesis, and nutrient metabolism [47]. Local stress induces H_2_O_2_ and Ca^2+^ signals, activating CPKs and CBLs-CIPKs to phosphorylate RbohD. RbohD generates H_2_O_2_, propagating Ca^2+^ signals via RLKs. Ca^2+^ and H_2_O_2_ signals mutually activate, triggering systemic responses. Calcium also activates SnRK2s and MAP kinase cascade via CPKs and CBLs-CIPKs, leading to phosphorylation of drought-related TFs (BZIP, WRKY, MYC, MYB) and activation of responsive genes. This induces ABA-dependent and ABA-independent responses, including ROS production. Antioxidative enzyme concentration increases, while protective molecules (osmolytes, aquaporins, LEA proteins) contribute to stress responses [48]. Stress signaling regulates ion and water transport, metabolic reprogramming, and gene expression for cellular stability. Primary hyperosmotic stress triggers ABA accumulation, eliciting adaptive responses in plants [49,50]. The drought stress induces gene expression, impacting the metabolism of various biochemicals such as enzymes, transcription factors, hormones, amino acids, and carbohydrates [51]. Key molecules including abscisic acid (ABA), proline, tryptophan, late embryogenesis abundant (LEA) proteins, trehalose, raffinose, mannitol, glycine-betaine, and superoxide dismutase are to be considered while defining strategic breeding programs for drought tolerance [52]. These compounds play roles in dehydration avoidance or tolerance, involving osmotic adjustment, membrane stabilization, antioxidation, reactive oxygen species (ROS) scavenging, and gene regulation [53].

### 3.3. Molecular Adaptation

The activation of multiple genes induces physiological and biochemical responses to drought stress, facilitated by modifications in cell wall proteins [54]. Water deficiency stress signaling involves proteins like transcription factors (DREB, WRKY, bZIP, bHLH, NAC, MYC, MYB), protein kinases (MAPK, CDPK), and receptors [55]. Plants utilize both ABA-dependent and ABA-independent pathways to sense and respond to drought stress, with ABA-independent TFs serving as molecular switches during signal transduction [56]. Another implicated mechanism in stress signaling is the increased generation of reactive oxygen species (ROS), associated with ABA and Ca^2+^ increases under drought stress [57]. Accumulation of ROS serves as a stress signal, and protective molecules like osmolytes, heat shock proteins, aquaporins, and LEA proteins play crucial roles in plant stress responses [48]. Stress signaling induces gene expression, leading to the synthesis of proteins responsible for various activities, including transcriptional regulation, cell membrane protection, antioxidant biosynthesis, and water and ion uptake [58]. The identification of the ‘ERECTA’ gene, regulating transpiration efficiency in Arabidopsis [59], transcriptional analysis of wheat genotypes with differing transpiration efficiency contribute to understanding the molecular basis of isotopic discrimination [60]. The sequential process from stress perception to the adaptive response in wheat under drought conditions is depicted in Figure 3.

**Table 2 plants-14-01053-t002:** Comprehensive analysis of phenotypic traits influencing key yield components across diverse drought-prone environments.

Trait	Environment	Yield Factor	Techniques	Reference
Seedling vigor	Pot, dry	Water uptake	Visual	[61]
Cooler canopy	Drought, field	Deep root	Visual	[62]
Leaf architecture	Any, dry	Water use efficiency	Visual/metric	[61]
Leaf rolling	Drought, field	Transpiration and water loss	Visual	[62]
Canopy green area	drought	vegetation index	Red, green, blue imaging	[63]
Leaf area index	drought	Normalized difference vegetation index	Red, green, blue imaging	[64]
Normalized difference vegetation index	Drought, field	Canopy temp	Red, green, blue imaging	[65]
Stay green traits	drought	-	Red, green, blue imaging	[66]
Semi-dwarf habit	Any	Harvest index	Visual/molecular markers	[67]
Root diameter	Laboratory, field	Associated with seed yield	Wax-layer screen	[68]
Deep roots	Field, dry	Water uptake	Infrared thermometry	[69]
Deep root	Laboratory, field	Greatest number of shallow roots	Wax-layer screen	[68]
Root architecture	Laboratory, field	Nitrogen uptake efficiency	High throughput laboratory screens	[70]
Growth rate/biomass	Any, Dry	Water uptake, water use efficiency	Metric/spectral reflectance	[71]
Biomass, leaf area index	Field	Green area indexes	Conventional digital cameras	[72]

An integrated approach for transpiration efficiency could involve physiological tests (D analysis), markers for QTLs controlling D, and genes like ERECTA for allelic variation or plant transformation [28]. Genes involved in cellular structure maintenance and protection are key targets for drought-tolerant crop development, with complex regulation in response to limited water availability. These genes encompass those affecting cell growth (mostly downregulated), hormone synthesis (ABA, proline metabolism, ROS-scavenging enzymes), carbohydrate metabolism (activated or upregulated), root-specific genes related to cell expansion, and protective proteins like late embryogenesis abundant (LEA) and chaperones. Additionally, genes encoding isopentyl transferase (IPT) contribute to delayed senescence, sustaining high photosynthetic activity during drought episodes [73,74] that can be important for breeding drought-tolerant wheat (Figure 3).

## 4. Breeding Strategies for Developing Drought Stress Tolerance in Wheat

### 4.1. Convetional Approaches

#### 4.1.1. Utilization of Wild Species for Trait Manipulation

Crop wild relatives (CWR) sharing genetic components with major crops, offer untapped potential for enhancing global food security. Even after transferring drought tolerance genes, introgression species are crucial for maintaining genetic diversity, combining multiple genes, and monitoring traits to ensure optimal performance of breeding lines under diverse conditions [75]. Utilizing gene donors like *Aegilops*, *Elymus*, *Lolium*, *Dasypyrum*, *Thinopyrum*, and *Triticum* enhances genetic diversity in bread wheat, leading to the development of genotypes resilient to abiotic stresses [76,77,78].

Integrating the wild gene pools into breeding programs ensures a continuous supply of genetic resources for desirable traits, contributing to effective crop improvement efforts [79,80,81,82]. The successful contributions of the wild gene pool to drought-tolerant traits and introgression of novel genes in wheat improvement via breeding techniques have been noted (Table 3). However, challenges arise in introgression, with hurdles like yield reduction from rye allelic transformation (e.g., Sr50). Trade-offs may occur when replacing segments or chromosomes, impacting agronomic features and potentially introducing non-selected traits and linkage drag. Despite difficulties, working with wild species provides opportunities to discover new traits.

#### 4.1.2. Backcross Breeding

Traditionally, backcrossing is a widely used technique to introduce one or a few desirable genes into an elite breeding line with mostly favorable traits [92]. Marker-Assisted Backcrossing (MABC) involves transferring desirable genes or QTLs from a donor parent into an elite recurrent parent using molecular markers. Unlike conventional backcrossing, MABC selects progeny based on marker alleles rather than phenotypic traits [93]. When phenotypic selection is challenging, marker-assisted backcrossing enhances breeding efficiency by selecting BC offspring with donor parent marker alleles near the target gene. Markers also minimize linkage drag by identifying progeny with reduced donor genome outside the desired region [94]. Three levels are used to complete the MABC [95]. The target gene or QTL is screened using markers at the first level, which is referred to as “foreground selection” [96]. Recessive alleles are screened via foreground selection, which takes a lot of time when using traditional techniques. Additionally, selection occurs at the seedling stage, avoiding the time-consuming phenotypic screening processes and enabling the selection of those plants that possess the desired gene [97]. Recombinant selection, the second stage of MABC, selects backcross progenies with recombination events between the loci of interest and flanking markers. Through this selection, the amount of introgression—that is, the donor chromosome carrying the target locus—is decreased. Even after several backcross generations (>10; ref. [98]), the donor’s chromosomal region is still substantial in traditional backcross breeding. However, in MABC, this donor chromosomal region (linkage drag) is decreased with the aid of flanking markers (e.g., <5 cM on each side of the target gene) [99]. Since double recombination events on both sides of the target locus are often uncommon, recombinant selection is typically carried out using two backcross generations [99]. Using genome-wide dense molecular markers, the third step of MABC, referred to as “background selection”, chooses backcross progenies with the largest genomic area of the recurrent parent [100]. Selection is made against the donor genomic region since these genome-wide dense markers are not connected to the target gene or QTL [99]. According to [99], background selection is therefore particularly helpful in speeding up the recovery of the recurrent parent’s genetic complement, which would otherwise take significantly longer (six or more backcross generations) using the traditional backcross approach. With MABC, the BC2 or BC3 generation’s recurrent parent genome is retrieved [96,100,101]. MABC efficiently transfers desired traits to an elite recurrent parent with minimal genetic alteration, making it a practical and cost-effective MAS approach [102]. This method works well for crops with low heritability features that are challenging to phenotype and choose using important quantitative traits, particularly those that are expressed later in the development stage [103]. However, considering that quantitative features are influenced by environmental factors, epistatic factors, and polygenes, transferring one gene at a time using MABC can be difficult [104]. Four drought tolerance quantitative trait loci (QTLs), canopy temperature, vegetative index, chlorophyll content, and grain yield, were transferred from the drought-tolerant donor line C306 to the well-known high-yielding, drought-sensitive variety HD2733 in [48] study. Rigorous phenotypic screening and marker-assisted selection advanced each generation, resulting in 23 enhanced lines with four drought-tolerance QTLs and background recovery rates of 85.35% to 95.79%. When subjected to moisture-deficit stress, the backcross-derived lines produced more than the recipient parent. The emergence of drought-tolerant HD2733 was a result of the four QTLs that were introgressed using MABC technique. Although wheat’s drought resistance has improved more quickly thanks to the MABC technique [105,106], effective MABC experiments for drought tolerance are still in the minority. Backcrossing high-yielding wheat lines with donors carrying drought resistance genes led to the development of the Indian wheat variety HD 2967, DBW 187, HI 1605, and HD 2733 [107].

#### 4.1.3. Mutation Breeding and TILLING

Wheat has undergone conventional mutation induction through physical methods like X-Rays and Gamma Rays as well as chemical treatments with diethyal sulphonate, ethyl methan sulphonate and targeting induced local lesions in genomes [108]. The TILLING technique enhances control over detecting mutagenized genome regions, enabling reverse genetics and non-transgenic resistance development. Polyploids like wheat are favorable for TILLING due to genome duplications [109]. A well-known example of Indian wheat variety development is Sharbati Sonora, which was created through gamma-ray-induced mutation breeding from the Mexican wheat variety Sonora 64, introduced to India during the Green Revolution [110]. There are several examples like the Njoro BW1 wheat varieties developed in Kenya, this variety emerged from gamma radiation-induced mutations in the wheat genotype ‘Kwale’. Njoro BW1 exhibits enhanced drought tolerance and is recommended for cultivation in Kenya’s marginal areas [111].

#### 4.1.4. Recurrent Selection

Recurrent selection is thought to be an effective method for pyramiding multiple traits in plants [112], its effectiveness in selection is unsatisfactory due to the fact that genotypic selection takes a long time (two to three cropping seasons for a cycle of selection) and phenotypic selection is dependent on the environment [93]. The goal of recurrent selection is to increase the frequency of favorable alleles in populations through cyclic selection, evaluation, and recombination. When markers are used, this process is known as Marker-Assisted Recurrent Selection (MARS), in which genome-dense markers associated with multiple favorable traits (gene/QTL) of interest from various sources are identified, and selection is then conducted based on genomic regions involved in complex trait expression to assemble the best genotypes in a population [61]. It allows selection at genotypic level and intermating for first selection cycle during the same crop season and hence improves efficiency and accelerates the conventional selection [113]. According to [114], MARS permits the phenotyping of F2 derived generations (such as F4 or F5), followed by the genotyping of F2 or F3 (for calculating marker effect) and two to three cycles of recombination based on the presence or absence of marker alleles for minor QTL. The base population, which is created by crossing superior lines, is used to identify QTL. Additionally, to gather alleles for a key QTL in a single background, lines with the best, superior, and necessary alleles are crossed. Superior lines for varietal development are chosen by screening derived lines from crossings based on phenotypic criteria. MARS is an improved system that enables genotype selection and intercrossing in one cropping season (Figure 4), which can facilitate the efficacy of recurrent selection and expedite the selection process [115] and help in integration of multiple favorable genes. There is one previous report on the successful use of MARS in wheat [116]. These authors recombined 4–8 favorable QTLs for yield, drought- and heat-adaptive traits discovered in the base population, and found progeny with superior grain yield compared with check cultivars and parents. Comparing MARS to MABC, several large and minor QTL are caught, resulting in greater genetic gain [117]. MARS has been used to improve a breeding population in relation to QTLs that have less of an impact on the phenotypic than MABC. Maize, soybean, sunflower, wheat, sorghum, and rice are among the crop species that MARS has been successful in increasing drought tolerance in [118,119,120,121]. Figure 4 depicts marker-assisted recurrent selection in wheat, highlighting the key adaptive strategies involved.

To create F1 offspring, individual parents from the population are chosen to cross. The single-seed decent technique is used to self-produce 300 F2 offspring from F1 individuals. F2 people progressed to F3, F4, and F5. Genotyping is carried out on individual progenies. The best seeds are selected and evaluated in multi-location phenotyping. Quantitative trait loci (QTL) analysis is performed and modeling is conducted prior to selection of QTLs for recombination. The six best genotypes per offspring of F_3_ are selected (A–F). In order to create F2, F2 are progressed to F3 and F4 for multi-location phenotyping, the F1 individuals from the first recombination cycle were selfed (adapted from [112]. ‘Kundan’ is an excellent example of an Indian wheat variety developed through recurrent selection [122]. By repeatedly selecting and interbreeding superior individuals, breeders were able to improve key traits such as grain yield, disease resistance, and adaptability.

### 4.2. Molecular and Metabolic Approaches

#### 4.2.1. Development of DNA Based Markers for Drought Tolerance in Wheat

Developing genomic resources for wheat drought tolerance involves sequencing the complex hexaploid genome, identifying candidate genes through bioinformatics, and employing techniques like QTL mapping and association mapping. Genomic selection utilizes DNA markers to predict genetic potential and validate markers, such as SNPs and SSRs, for specific drought-related genes or regions. These markers enable MAS, streamlining the breeding process for rapid and precise trait selection. Drought-tolerant wheat lines are validated through field trials under stress conditions to ensure adaptability. The process of developing genomic resources is illustrated in Figure 5.

#### 4.2.2. Marker-Assisted Selection

Marker-Assisted Selection (MAS) using DNA markers associated with agronomic traits has addressed breeding gaps, with over 150 functional genes and QTLs identified in wheat. For a relatively quick, precise, and straightforward assessment of genetic diversity, molecular markers such as microsatellites, amplified fragment length polymorphism (AFLP), and random amplified polymorphic DNA (RAPD) are quite effective [123]. Molecular markers can help detect novel genomic regions associated with drought tolerance and facilitate Marker-Assisted Selection (MAS) by focusing on the most promising candidate genes [124]. According to [125], microsatellite markers, also referred to as simple sequence repeat (SSR10) markers, are among the best options for wheat due to their high polymorphism rate, co-dominant character, multi-allelic composition, chromosomal specificity, and extensive distribution throughout the genome. A few markers can show contrasting relationships between small to large populations of wheat genotypes due to their distinctive features [126]. Utilizing microsatellite markers to assess the genetic diversity of wheat germplasms for drought-tolerance breeding programs has been strongly advised by recent findings [125,127,128]. Both the microsatellite marker-based and morpho-physiological trait-based approaches for evaluating germplasm have been widely applied, either separately or in combination, to wheat populations on several continents. DNA markers linked to recognized drought genes may be helpful for quickly screening a large number of genotypes with minimal expense and effort.

Figure 6 presents a graphical representation of gene pyramiding for sustainable crop improvement against abiotic stresses. By integrating two or more complementary genes, Marker-Aided Gene Pyramiding (MAGP) improves trait performance and corrects deficiencies by introducing genes from donor sources, increasing the longevity of disease resistance [99]. With the advent of molecular breeding, further new approaches, such as genomic selection and MARS, are developed for overcoming the limitations of MAS and MABC, particularly when multiple QTLs control the expression of complex traits (Figure 6).

The DBW 88 is an excellent example of the successful use of Marker-Assisted Selection (MAS) in Indian wheat breeding [129]. MAS allowed breeders to precisely introduce and track beneficial genes like Lr24 and Yr15 for disease resistance. Its wide adaptability, disease resistance, and ability to perform under heat stress make it an ideal choice for wheat farmers in India, particularly in the North Western Plains Zone.

#### 4.2.3. Quantitative Trait Loci (QTLs), MQTLs, and e-QTLs

A QTL is a site where certain genes affect a quantitatively inherited trait’s phenotype. QTL mapping (polygenes) can be used to investigate a crop’s genetic variability [129]. Estimating the locations, magnitude, and quantity of impacts on the phenotypic and gene activity pattern is made possible by QTL mapping [130]. Ref. [131] conducted the first research on cloning QTLs for drought resistance traits, with further insights provided [72]. In wheat, due to drought stress, the place of genes which had an influence on abscisic acid (ABA) concentration was detected [132] in 5A chromosome transports gene(s). QTLs for resilience to drought were mapped in hexaploid wheat on chromosomes 1A, 1B, 2A, 2B, 2D, 3D, 5A, 5B, 7A, and 7B [133]. Recombinant inbred lines from crossing of drought-resistant and drought-susceptible cultivars were used to create mapping populations for QTL analysis regulating yield under drought [134].

QTL mapping, while impacted by various factors, is enhanced through multi-QTL (MQTL) analysis, which improves selection accuracy and efficiency in breeding programs. Meta-analysis of QTLs, combining multiple studies, reduces confidence intervals, providing deeper insights into trait-related regions. Successful applications of MQTL analysis have been reported in rice, soybean, cacao, potato, barley, cotton, and wheat (Table 4). Expression quantitative trait loci (eQTLs) are associated with genomic regions and gene expression levels, highlighting genetic variants that influence gene regulation. eQTL analysis, enabled by high-throughput technologies such as microarrays and RNA sequencing, helps to understand the genetic basis of complex traits like drought tolerance. Investigating eQTLs specific to drought stress in wheat unveils the genetic factors for designing the plant’s response to water scarcity, thereby aiding the development of resilient wheat varieties in breeding programs. Specific QTLs for grain yield under heat and drought stress conditions were identified and pyramided into the Indian wheat variety HI 1544 [135]. By using QTLs associated with stress tolerance, the breeders enhanced the plant’s ability to maintain productivity under high temperatures and water stress, a critical factor in many wheat-growing regions in India.

#### 4.2.4. GWAS, Genetic Linkage Mapping, and Transcriptomics for Drought Tolerance

GWAS/MTA surpass the QTL mapping limitations, leveraging genome-wide SNP or DArT markers to identify trait associations in a diverse wheat germplasm. Challenges arise from the large, complex wheat genome and incompatible sequence but reference genome availability aids in annotating functional genes. This enhances understanding of genome architecture and the relationship between drought tolerance genes/QTLs and conditioning factors. Plant survival strategies involve adapting metabolic activity and growth rate to mitigate stress damage, utilizing tactics for environment-specific survival. Genes encoding the dehydration response element-binding protein (TaDreb-B1) are among the most important modulators of intricate genetic networks under drought stress in wheat.

#### 4.2.5. Phenomics and Metabolomics

Phenotype manifestation due to environmental influences. Developing higher yielding crops requires documenting physiological phenotypes beyond genomics. The difficulty in accurately phenotyping plants, except for obvious traits, has been a longstanding limitation. To address this challenge, ref. [143] proposed a molecular breeding framework for drought tolerance based on the Passioura equation, incorporating water use, water use efficiency, and harvest index.

Metabolomics, studying small molecules (metabolites), is emerging as a diagnostic tool for plant performance in breeding and crop improvement. Metabolic markers, increasingly replacing traditional molecular markers, provide insights into the interaction between biological systems and their environment, aiding in the selection of improved breeding materials. The production of secondary metabolites and their function is depicted in Figure 7. Plants produce primary metabolites for essential functions and secondary metabolites (SMs) with defensive and signaling roles under stress conditions (Table 5). Understanding the biosynthetic pathways of plant secondary metabolism enhances knowledge of SMs in stress physiology (Figure 7). Drought stress induces changes in carbohydrate metabolism, leading to the accumulation of osmolytes and organic acids, causing productivity loss. Glycolysis and the tricarboxylic acid cycle (TCA cycle) are key metabolic pathways, but, under stress, primary metabolism shifts to secondary metabolism, synthesizing various small organic molecules, i.e., SMs. Plant SMs, grouped into phenolics, terpenes, and nitrogen-containing compounds, play crucial roles in stress response (Figure 7).

Metabolomic studies in plants aim to identify and quantify primary metabolites essential for normal growth and secondary metabolites crucial for survival under adverse conditions. The plant metabolome, comprising around 30,000 endogenous metabolites, includes carbohydrates, amino acids, organic acids, lipids, plant hormones, and signaling molecules, playing vital roles in growth and development [144]. Specialized plant metabolites (SPMs), or secondary metabolites, play vital roles in plant responses to various stresses. Two main approaches to enhance SPM production involve leveraging genetic variability and employing breeding and biotechnological strategies. Assessing crop diversity using molecular markers can link genetic variations to specific SPM content. Agronomic management practices, such as soil fertilization and interactions with beneficial microorganisms, also influence SPM levels. Flavonoids, acting as signaling molecules and antioxidants, contribute to stress tolerance. Identifying genes and loci responsible for SPM accumulation through quantitative genetic approaches aids in breeding for stress tolerance.

**Table 5 plants-14-01053-t005:** Advanced analytical protocols for identifying metabolites altered in wheat in response to drought stress.

Annotated Metabolites	Sample	Analytical Platform	References
Targeted and non-target analysis: amino acids, organic acids, sugars, sugar alcohols, and organic antioxidants	Root and leaf tissue	GC-MS	[145]
Non-target analysis: amino acids, organic acids, sugars, polyols, glycolysis cycle, and GABA shunt metabolites	Shoot tissue	TOF	[146]
Non-target analysis: amino acids, organic acids, sugars	Leaf tissue	GC-MS	[147]
Non-target analysis: lipids, sugars, oxidative stress compounds, and phytohormones	Root tissue	RPLC-Q-TOF	[148]
Increase in total phenolics, flavonoids, anthocyanins	Leaf tissue	Colorimetric method	[149,150]
Induction in sugars, amino acids, organic acids	Leaf tissue	GC–MS	[146,151]
Tannins	seeds	RT-PCR	[152]
Flavonoid	Leaf tissue	Colorimetric method	[150]
AA (serine, asparagine, methionine, lysine)	Seeds	GC/MS	[147]
Organic compounds, phenols, flavonoid	Leaf tissue	GC/MS	[17]
Soluble sugars and proline, proteins, inorganic solutes	Seeds	Biochemical methods	[153]
Proline, protein content, total soluble sugars	Leaf tissue	Biochemical methods	[154]

GC-MS = gas chromatography mass spectrometry, C-TOF = time-of-flight mass spectrometry technique, RPLC-Q-TOF = high pressure liquid chromatography coupled with quadrupole time-of-flight mass spectrometry, RT-PCR = real-time reverse transcriptase–polymerase chain reaction.

While metabolomics provides valuable insights, experimental design considerations, including genotype, developmental phases, and biological replicates, are crucial for effective utilization in breeding programs [155]. Plant metabolomic research relies on evolving methodologies and instrumentation for comprehensive metabolite identification, quantification, and localization. The key strategies include: *metabolite profiling*, involving the identification and quantifying predefined metabolites linked to specific metabolic pathways; metabolic fingerprinting, involving screening samples to distinguish biological variations or origins; metabolite target analysis, involving qualitative and quantitative analysis of one or a few metabolites related to a specific metabolic reaction; metabonomics, involving the analysis of tissues and biological fluids for changes in endogenous metabolite contents due to diseases or therapeutic treatments. Integrated mass spectrometry-based methods make metabolomics a robust technology, showing potential in plant metabolomic studies across various crops [144]. Drought induces oxidative stress in plants, resulting in reactive oxygen species (ROS) production. Flavonoids and polyphenols, common natural compounds, help plants scavenge ROS, reducing membrane lipid peroxidation and strengthening cell walls. These SMs enhance the TCA cycle, glycolysis, and the glutamic acid-mediated proline biosynthesis pathway, crucial for osmotic regulation. Metabolomics reveals drought-induced changes in plant metabolism, with wheat showing increased amino acids, sugars, and organic acids. Flavonoids and polyphenols also differentially accumulate in drought-resistant and sensitive cultivars. In wheat, spikes exhibit higher flavonoid accumulation and greater tolerance to drought than flag leaves. Additionally, under drought stress, lignin deposition in wheat is enhanced by syringaldazine peroxidase, impacting the polyamine degradation process.

### 4.3. Genomics-Assisted Breeding Approaches

Genomic-assisted breeding enhances traditional methods by utilizing genetic markers like SNPs and SSRs to track and select specific traits. Marker discovery and genetic mapping enable the association of markers with a trait of interest. This accelerates trait selection, reducing the reliance on time-consuming phenotypic screening. Genomic selection offers a comprehensive approach, estimating genetic improvement for multiple traits simultaneously, and is particularly useful for complex traits. Success of genomic-assisted breeding relies on accessible genomic resources, including reference genomes and marker databases, facilitated by advances in DNA sequencing and bioinformatics. Genomic-assisted breeding has the potential to revolutionize agriculture, addressing food security, environmental sustainability, and economic challenges by improving breeding program speed and accuracy (Table 6).

#### 4.3.1. Genomic Breeding Approaches for Designing Drought Tolerance in Wheat

Drought-tolerant wheat varieties are developed more quickly due to genomic breeding methods such as haplotype breeding, genome editing, Marker-Assisted Backcrossing (MABC), Marker-Assisted Gene Pyramiding (MAGP), and Marker-Assisted Recurrent Selection (MARS). Marker-Assisted Gene Pyramiding (MAGP) involves simultaneously introgressing multiple drought tolerance genes or genomic regions into a wheat variety using associated markers, enhancing resilience to drought. The complexity of gene expression dynamics and regulation at the subgenomic level in wheat, including alternative splicing and epigenetic modifications, influences the stacking of drought tolerance traits. Sub-genome biased alternative splicing persists in polyploid wheat, impacting gene expression balance. Epigenetic modifications, such as histone methylation and acetylation, play crucial roles during wheat evolution and embryogenesis, affecting gene regulation and chromatin accessibility. Understanding these mechanisms provides opportunities for fine-tuning the expression of multiple drought tolerance traits in wheat breeding. QTL mapping, despite limitations in allelic diversity and genomic resolution, provides insights into genomic regions controlling specific traits, aiding in wheat improvement for drought tolerance. Combining different techniques accelerates the development of drought-tolerant wheat varieties and is contingent on comprehensive genomic resources and regulatory considerations.

#### 4.3.2. Genomic Selection

Genomic selection predicts wheat lines’ genetic potential for drought tolerance using genome-wide markers. Concurrent selection of highly saturated genome-dense markers is desirable, in which some are expected to be in linkage disequilibrium with all genes in a genome [156]. The fundamental prerequisites for genomic selection are high throughput markers, new statistical tools, and extremely effective computational techniques. Values known as genomic estimated breeding value are used to select stress-resilient lines [157]. Based on genome-dense, highly saturated markers, the genomic estimated breeding values (GEBVs) are estimated values derived using new statistical techniques [158]. In genomic selection, statistical models such as Bayes regression, Ridge, and best linear unbiased prediction (BLUP) will aid in predicting the genomic estimated breeding value [157]. Individuals from the training population are first genotyped and phenotyped, after which the data are statistically analyzed. Then, genomic estimated breeding values are computed and selection of best individuals is performed to develop a breeding population. Additionally, rather than using QTL effects, marker effects are used to determine the genomic predicted breeding values [158,159].

#### 4.3.3. Haplotype

The process of haploid production is depicted in Figure 8, illustrating the key steps involved in its methodology. According to [160], a haplotype is a collection of alleles for distinct polymorphisms (such SNPs, insertions/deletions, and other markers or variations) found on the same chromosome that are inherited together with little likelihood of modern recombination. A particular length of chromosomal DNA contains two haplotypes for each person, but the same stretch might have several haplotypes at the population level [161]. In other words, a haplotype is defined as a set of nearby genomic structural variations, such as polymorphic SNPs, with strong linkage disequilibrium (LD) between them [162]. Haplotype breeding in wheat under drought stress involves selecting and developing wheat varieties with improved tolerance to water scarcity. This approach leverages the power of genomics to accelerate the development of drought-resistant wheat varieties tailored to specific environmental challenges. Fast-growing capacity and affordability of DNA sequencing has motivated large-scale germplasm sequencing projects, thus opening exciting avenues for mining haplotypes for breeding applications [162]. Haplotype breeding targets specific combinations of alleles associated with drought tolerance, capturing synergistic effects which can play an important role in tailoring drought-tolerant wheat varieties.

**Table 6 plants-14-01053-t006:** Genomic-assisted breeding approaches for drought tolerance in wheat.

Traits	Chromosomes	Study Approach	Population Type	Reference
Drought susceptibility index	-	RAPD	wheat genotypes	[163]
DH, PH, TKW	12A	SSR and RFLP	Back cross	[164]
Normal and drought stress	-	SSR Marker	RILs	[165]
Normal and drought stress	-	SSR Marker	DH	[166]
Normal (ramandi 2014)	-	SSR and DArT	RILs	[167]
NDVI and grain yield	2A, 2D, 3B, and 5A	SSR/MABC	HD2733 × C306 (donor)	[168]
Grain yield and other traits	2A, 3B, 4B	MABC	BC_1_F_1_	[105]
Grain quality and rust resistance	-	MABC	F_2_–F_5_	[9]
Grain yield	-	MABC	F_3_ to F_8_	[168]
Root length, root weight, root density, root diameter	2A, 2B, 2D, 5B	Marker–trait association	Core collection	[169]
NDVI, GFD, TKG, grain yield	-	MARS	BC_1_/BC_2_ F_2_	[116]
Grain yield and biological yield	1D, 4A	MARS	Backcross populations	[170,171]
NUE and photosynthesis	1B and 5A.	Haplotype breeding	Wheat cultivars	[172]
DH, PH, and TKW	3D, 4A, 5B, 7A, and 7B	GWAS	Diverse wheat genotypes	[173]
Grain yield	-	Genomic Selection	Wheat lines from CIMMYT	[174]
Root growth angle	DRO1-like genes	Genome editing	NARC 2009 and Galaxy variety	[175]
YLD, PH, TNPM, TKG, GNS, SL, HI	1A, 1B, 1D, 2B, 3A, 3B, 6A, 6B and 7A	Genetic linkage mapping	RILs	[175,176]
NDVI, CT, PH, GWPS, TKG and YLD	2A, 5D, 5A and 4B	Genetic linkage mapping	RILs	[177]
Root length and root weight	1B, 2A, 2B, 2D, 3D, 4A, 4B, 5A, 5B, 6A, 6B, 6D, 7A	GWAS	Core collection	[178]
Root numbers, root weight, seed weight, seed length	1B, 2A, 2B, 3B, 5A, 5B, 6A, 7A	GWAS	Landraces	[179]
Flavonoid biosynthesis	-	Transcriptomics	Wheat cultivars	[180]
Dehydrins and aquaporins	-	Transcriptomics	Landraces	[181]
PH, TNPM, DH, juvenile growth habit	2B, 4D	Exome sequencing	RILs	[62]
Aquaporins, LEA proteins	5B, 6D, 6B, 2B	RNA Seq	Landraces	[182]
RWC, TKW, awn length, coleoptiles length, shoot length	2A	QTL analysis	Core collection	[183]
Grain yield	3BL	QTL analysis	DH (RAC875 × Kukri)	[184]
YLD, CT, potential quantum efficiency, chlorophyll content	1A, 1D, 2B, 3A, 3B, 4B, 4D, 5B, 6A	QTL analysis	RILs (C306 × HUW206)	[139]

DH = Heading days, PH = Plant height, YLD = Grain yield, CT = Canopy temperature, RWC = Relative water content, TKW = Thousand kernel weight, TNPM = Tiller number per meter, NDVI = Normalized difference vegetation index, GWPS = Grain weight per spike, GNS = Grain number per spike, SL = Spike length, HI = Harvest index, GFD = Grain filling duration, QTL = Quantitative trait loci, GWAS = Genome wide association study, MARS = Marker assisted recurrent selection, MABC = Marker assisted backcrossing, DArT = Diversity array technology, RFLP = Restriction fragment length polymorphism, RAPD = Random amplified polymorphic DNA, SSR = Simple sequence repeat, DH = Doubled haploid, RILs = Recombinant inbred lines, BC = Backcross, F = Filial generations.

#### 4.3.4. Genome Editing and Sequencing

Genome editing techniques allows precise modification of drought-related genes. Scientists have been able to create specific changes in organisms of interest during the last ten years thanks to genome editing. The genetic engineering approach efficiency in organisms through genome editing involves nucleases, zinc-finger nucleases (ZFNs), transcription activator-like-effector nucleases (TALENs), and the clustered regularly interspaced short palindromic repeat (CRISPR/Cas systems) such as CRISPR/Cas9, CRISPR/sgRNA, and CRISPR/Cpf1. Notably, by incorporating genes from wild relatives exhibiting drought tolerance, CRISPR/Cas9 may facilitate the quick enhancement of drought and regionally adapted cultivars. As a result, new commercial types will be produced that preserve the stress resilience characteristics of their wild counterparts [185]. To increase wheat’s flexibility, foreign genes for heat and drought tolerance can be effectively transferred within or across genera and species. For instance, wheat adaptability and performance can be enhanced by the transfer of foreign genes that modulate stress-adaptive features, such as hormones, dehydration-responsive element-binding proteins (DREB), enzymes, and deeper rooting genes (DRO1) [175]. The DRO1 and DREB genes are two instances of intragenus transfer of a gene or genes. DRO1 affects the orientation of the root system in wheat and regulates the root development angle in rice [175]. Exone sequencing can be a useful method in researching the genetic basis of characteristics like drought tolerance in crops like wheat. DREB are implicated in tolerance to a variety of abiotic stimuli, including drought, salt, low temperature, and abscisic acid (ABA) in wheat [186]. By focusing on the protein-coding regions of the genome, researchers can identify genetic variations that may be associated with traits of interest. Exone sequencing involves the collection of tissue samples from wheat plants that exhibit either drought tolerance or susceptibility. Subsequently, DNA extraction is performed on these samples. Following that, exome capture techniques are used to specifically enhance and sequence the protein-coding sections of the wheat genome. This entails the development of probes or primers that selectively bind to exonic sequences. High-throughput sequencing techniques are used to sequence the exonic portions of the wheat genome that have been captured. Subsequently, the sequencing data are analyzed utilizing bioinformatic techniques in order to detect genetic variants, namely single nucleotide polymorphisms (SNPs) and tiny insertions/deletions (indels), inside the exonic regions. A comparison is conducted to analyze the genetic differences between drought-tolerant and drought-susceptible wheat types. Once variations have been identified, researchers must analyze their possible effects on gene function and their correlation with disorders. This involves comparing the variants to existing genomic databases and the literature to understand their significance. The goal is to find potential genes that are involved in drought resistance. These genes may possess variations that are more often seen in drought-tolerant cultivars. To confirm the involvement of the discovered candidate genes in drought tolerance, it is recommended to carry out further investigations, such as gene expression studies or functional tests. Gene differences have been found, which generate molecular labels that are linked to them. These markers are applicable in breeding programs for the purpose of selecting drought-tolerant characteristics. Breeding initiatives for wheat should include exome sequencing data to create new varieties with better drought tolerance. Exome sequencing provides a focused and cost-effective approach to identify genetic variations associated with specific traits. This information can be invaluable for crop improvement programs aiming to develop more resilient and drought-tolerant wheat varieties. Integrating genomic data into breeding efforts allows for more targeted and efficient development of crops with desired traits, contributing to food security and agricultural sustainability in the face of environmental challenges like drought. RNA sequencing, also known as RNA-Seq, is a powerful technique used to study and analyze the transcriptome of a biological sample. The transcriptome refers to the complete set of RNA molecules, including messenger RNA (mRNA), non-coding RNA, and other types of RNA, present in a cell or a population of cells at a specific time. RNA-Seq provides a high-throughput and quantitative method for profiling gene expression, identifying alternative splicing events, discovering novel transcripts, and studying the abundance of non-coding RNAs. By analyzing the transcriptome of wheat plants under drought stress conditions, researchers can identify key genes and pathways involved in the plant’s response to water scarcity. A study conducted by [182] revealed the role of aquaporins, ABA-signaling, dehydrins and late embryogenesis abundant proteins by performing RNA sequencing of the drought-responsive transcripts. Recent field research examining how drought affects wheat transcriptome alterations across reproductive stages found more than 300 genes that are differently expressed and implicated in several important activities, including stomatal movement, photosynthetic activity, and floral development [187]. In wheat, the AP2/ERF transcription factors are divided into four sub-families: RAV, AP2, ERF, and DREB [188]. ERFs have been the focus of several overexpression experiments to evaluate their potential for enhancing drought tolerance since they are quickly increased in response to stress [189]. The up-regulation of WRKY transcription factors at the protein level in response to drought stress indicates a direct involvement for these transcription factors in drought tolerance [190]. Under drought stress, the wheat WRKY transcription factors TaWRKY44 and TaWRKY93 were found to be essential response factors [191]. It is well known that zinc finger proteins (ZFPs) with a conserved QALGGH domain are linked to changes in gene expression during drought stress [192]. The expression of multiple genes encoding antioxidant enzymes, photosystem components, and enzymes representing carbohydrate metabolism and the tricarboxylic acid cycle was modulated in a pale green durum wheat mutant under drought stress, according to transcriptomic and proteomic analyses. These findings may be useful in addressing drought resistance in wheat [193].

#### 4.3.5. Bioinformatics and Speed Breeding

The wheat genome, housing 124,201 protein-coding genes across 21 chromosomes, serves as a model for polyploidy evolution, domestication, and genetic diversity studies [194]. Bioinformatic tools like SnpHub, GeneTribe, IGTminer, and Triti-Map facilitate wheat functional genomic research. A statistical model-based method, ggComp, leverages genomic variations to identify pedigrees in wheat germplasm resources. The integration of 1367 heat- and drought-responsive unigenes from the wheat HarvEST database with SNP markers using Biomercator V3.0 identified 137 SNPs on the consensus map [138].

Candidate gene sequences are used to BLAST 2.15, against the ‘Chinese Spring’ 5x genome [195]. Biomercator V3.0, based on Goffinet and Gerber’s algorithm, is used to conduct meta-analysis [196]. Multiple databases play a crucial role in wheat omics studies, as highlighted in Table 7.

Speed breeding accelerates wheat generation advancement, allowing multiplication of up to six generations in a year. This approach involves manipulating photoperiods in a controlled environment, resulting in quicker anthesis and seed ripening compared to traditional methods. Speed breeding facilitates accurate and rapid phenotype scoring, offering a powerful tool for integrating gene-edited wheat into breeding pipelines. Combined with other novel technologies, this approach addresses crucial limitations of developing sustainable and climate-smart wheat in shortest timeframe, contributing to global food security and the potential for a second ‘Green Revolution’ [197].

## 5. Future Road Map

Effective breeding for drought-tolerant, high-yield wheat necessitates the integration of traditional breeding, genomics, and physiological understanding. Marker-assisted selection based on DNA molecular markers complements trait-based selection. Despite climate change-induced recurrent drought affecting global wheat production, collaborative efforts among breeders and interdisciplinary experts are crucial [198]. Utilizing technologies like high-throughput phenotyping, next-generation sequencing, and genetic engineering can significantly enhance drought tolerance in wheat [199]. Transgenic approaches, utilizing genes like ‘*HVA1*’ from barley, have shown improved water use efficiency, biomass accumulation, and root weight in water-stressed wheat lines [200]. Similarly, a proline-inducing gene (P5CS) enhances drought tolerance in transgenic lines, potentially through proline’s antioxidant protection against oxidative damage [201]. Recognizing the association of drought with other stresses, and leading to simultaneous improvement of multiple stress factors, is essential for achieving better grain yield and quality in water-limited conditions [202].

Efficient mining and deployment of functional genes are crucial for wheat genetic improvement. Introgression populations from wild relatives into bread wheat have been developed, aided by genome assemblies and efficient tracking of introgression fragments [203]. Direct identification of superior haplotypes in wild diploid species, combined with marker-assisted selection, offers a strategy to avoid inter-subgenome interactions in cultivars [195]. Genomic data enable various methods like RenSeq, k-mer-based markers, and MutIsoSeq for cloning causal genes, providing potential tools for breeding drought-tolerant wheat [195].

## Figures and Tables

**Figure 1 plants-14-01053-f001:**
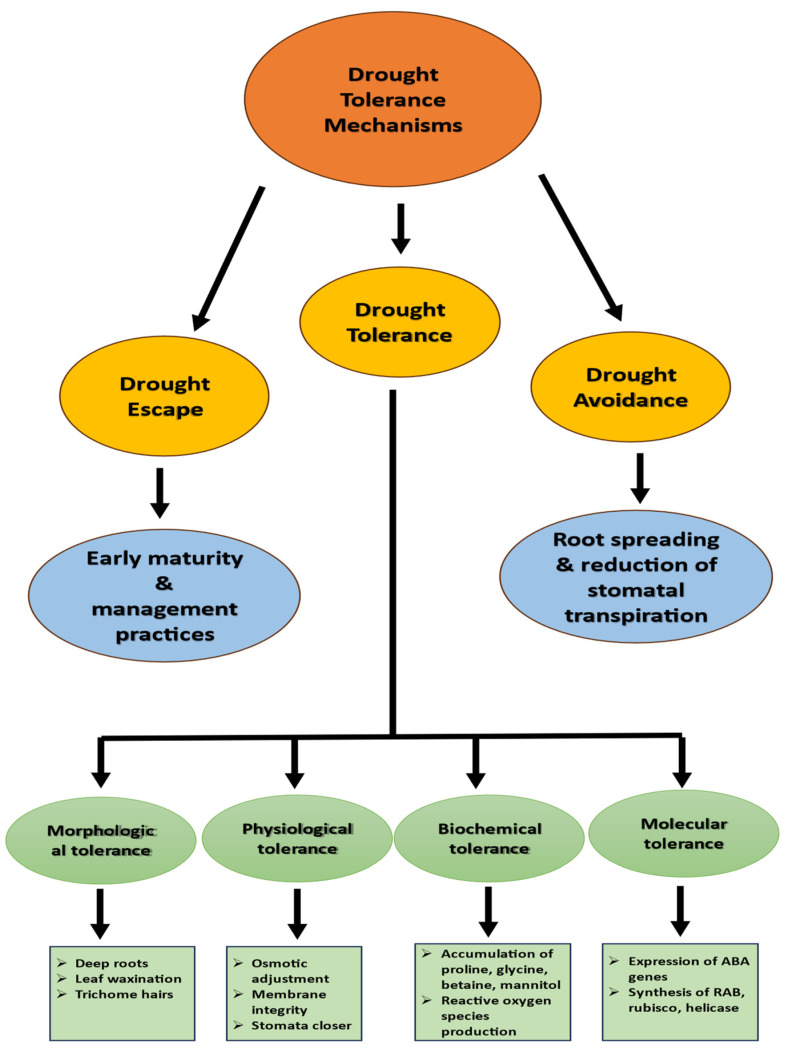
Drought tolerant mechanisms in wheat. This figure illustrates key adaptations that help wheat withstand water stress. It highlights morphological traits (deep roots and leaf wax, etc.), physiological responses (stomatal regulation and osmotic adjustment, etc.), and biochemical/molecular mechanisms (antioxidant activity, drought-responsive genes, ABA signaling). The figure visually summarizes these mechanisms to support researchers and breeders in developing drought-resilient wheat varieties.

**Figure 2 plants-14-01053-f002:**
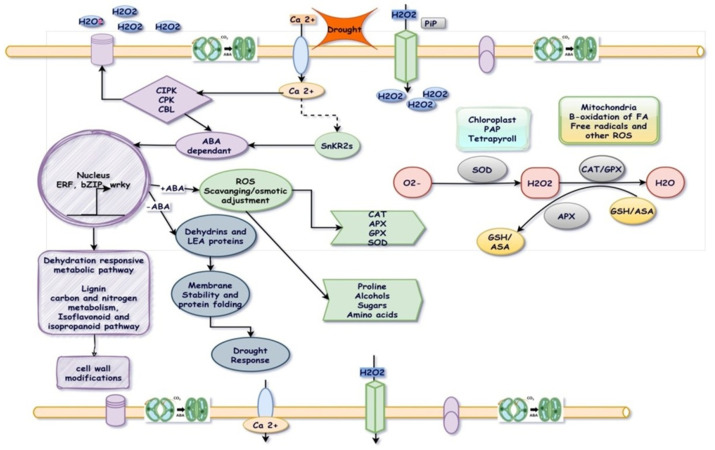
Wheat’s cellular and molecular mechanisms for drought resistance. This figure illustrates key pathways involved in drought adaptation. Arrows represent activation; bars indicate inhibition; and dashed lines depict postulated regulation. It highlights signaling pathways, gene expression, osmotic adjustment, antioxidant defense, and stress-responsive proteins. The figure aims to provide a clear visual representation of these mechanisms, aiding researchers in understanding regulatory networks for improving drought resilience in wheat.

**Figure 3 plants-14-01053-f003:**
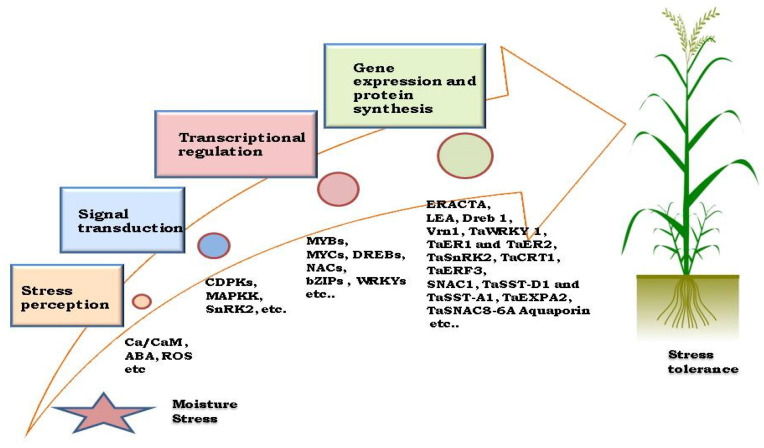
Stepwise response to drought stress in wheat depicts a flow diagram outlining the sequential process from stress perception to plant response. It includes stress perception (root and leaf sensors), signal transduction (hormonal signaling, ABA pathway), gene expression (activation of drought-responsive genes), and plant response (stomatal regulation, osmotic adjustment, antioxidant defense). The figure aims to visually summarize these interconnected processes, aiding researchers in understanding wheat’s adaptive mechanisms for drought resilience.

**Figure 4 plants-14-01053-f004:**
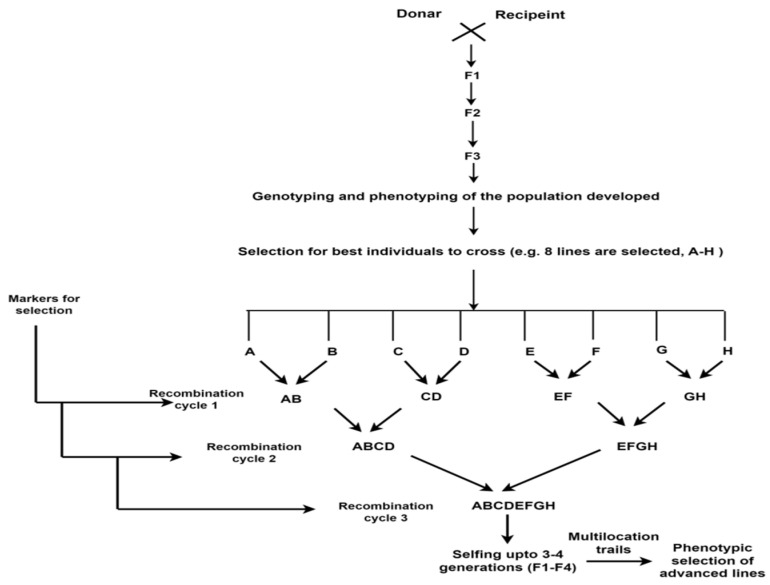
Schematic representation of Marker Assisted Recurrent Selection (MARS) illustrates the stepwise process of improving complex traits through marker-assisted selection. It depicts initial population selection, genotyping and trait evaluation, recurrent crossing of selected individuals, and advancement of superior lines over multiple cycles. The figure visually explains how MARS accelerates genetic gain, aiding breeders in developing high-yielding, stress-tolerant wheat varieties efficiently.

**Figure 5 plants-14-01053-f005:**
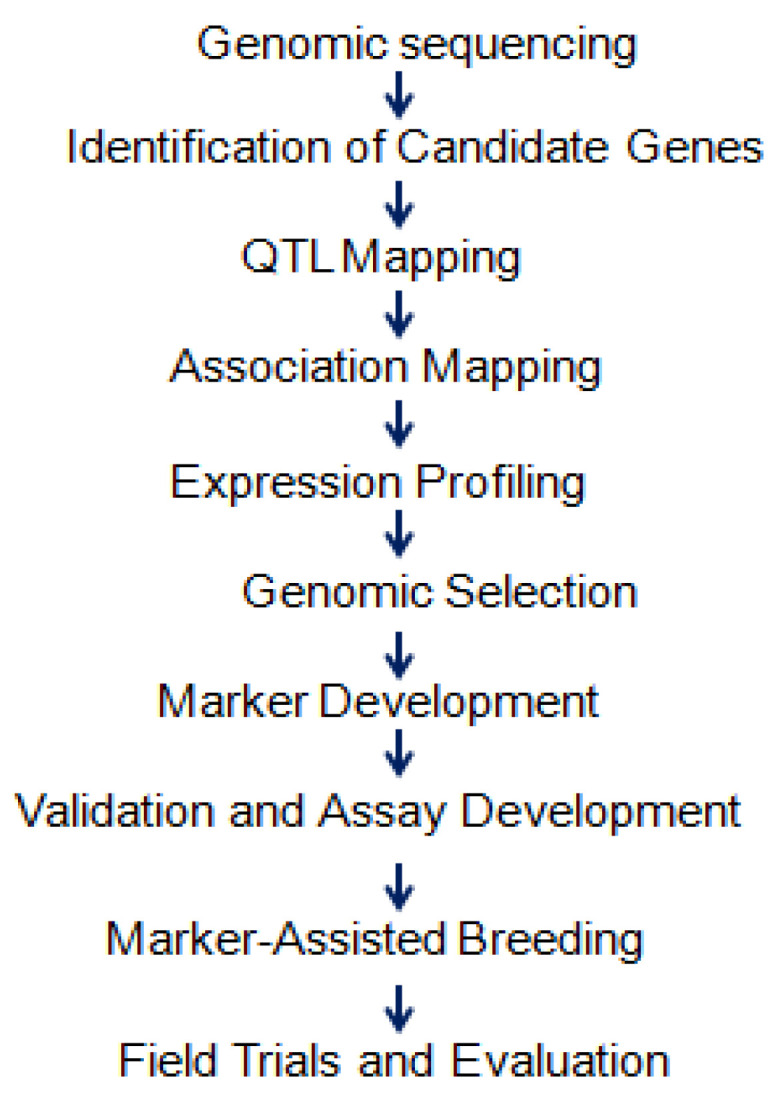
Process of development of genomic resources illustrates the stepwise approach to generating genomic tools for crop improvement. It includes DNA sequencing, marker discovery, genome assembly, genetic mapping, functional annotation, marker development, and application in breeding programs. The figure visually represents how genomic resources are developed to enhance genetic studies, aiding researchers and breeders in accelerating trait mapping and crop improvement efforts.

**Figure 6 plants-14-01053-f006:**
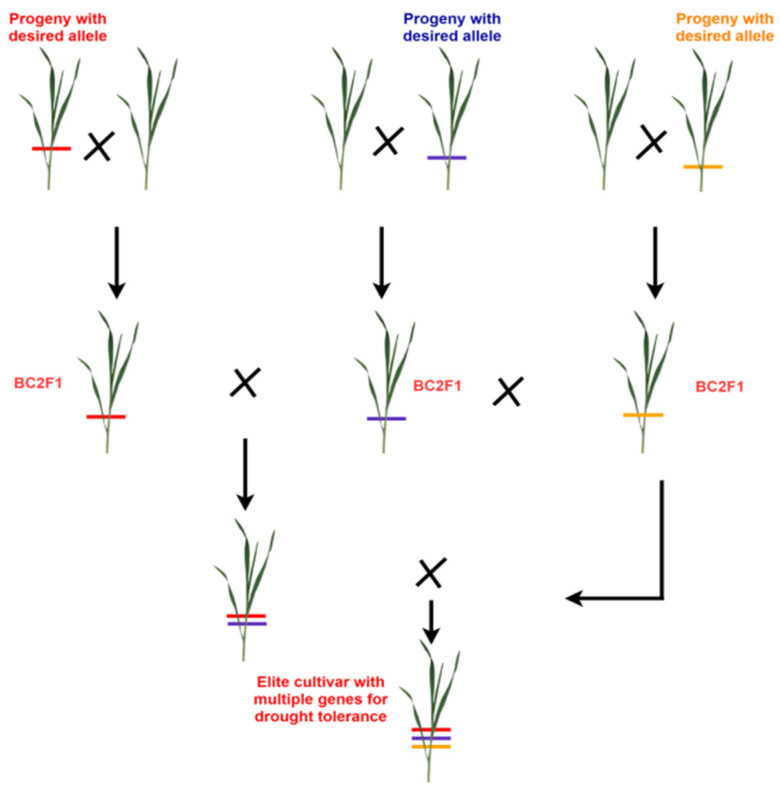
Graphical representation of gene pyramiding for sustainable crop improvement against abiotic stresses, which illustrates the strategic stacking of multiple stress-tolerant genes into a single genotype. It depicts gene identification, marker-assisted selection, and successive breeding cycles, enhancing resilience to drought, heat, and salinity. The figure visually conveys the process of gene pyramiding, aiding breeders in developing robust crop varieties for sustainable agriculture.

**Figure 7 plants-14-01053-f007:**
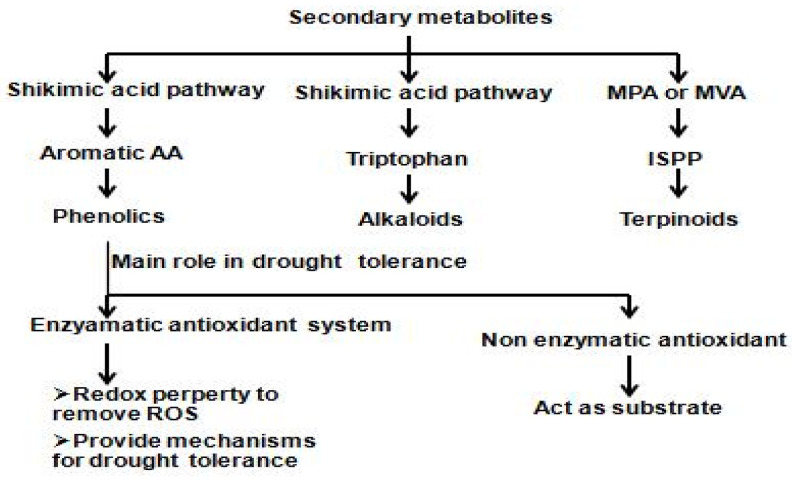
Pathways of production of secondary metabolites and their function, illustrating the biosynthetic routes leading to key secondary metabolites such as phenolics, alkaloids, and terpenoids, highlighting their roles in stress tolerance, defense, and growth regulation. The figure provides a visual overview of these metabolic pathways, aiding researchers in understanding their significance in plant adaptation and resilience.

**Figure 8 plants-14-01053-f008:**
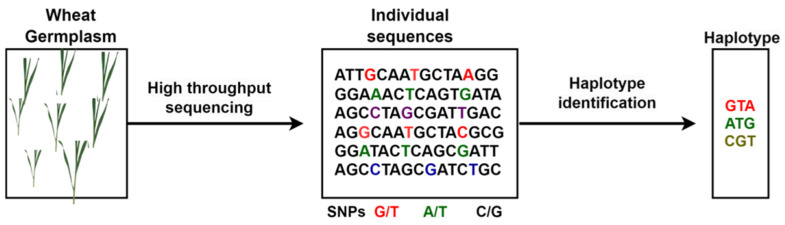
Schematic representation of haplotype production, illustrating the process of identifying and assembling haplotypes for genetic analysis. It depicts marker identification, linkage disequilibrium analysis, and haplotype block formation, aiding in trait mapping and selection in breeding programs. The figure visually summarizes haplotype production, supporting researchers in improving crop genetics and breeding efficiency.

**Table 1 plants-14-01053-t001:** Suitable trait combinations for different drought scenarios, Source [15].

Trait	Early-Season Drought (Pre-Anthesis)	Terminal Drought (Post-Anthesis)
Early vigor	˄	˅
Peduncle length	˄	˅
Relative water content	˄	˄
Leaf area index	˄	˅
More number of tillers	˄	˅
Low number of tillers	˅	˄
Tall size	˄	˅
Semi-dwarf	˅	˄
Early flowering and maturity	˅	˄
Prolonged—or short but high rate—grain filling	˅	˄
Flag leaf area	˅	˄

˄ = increased, ˅ = decreased.

**Table 3 plants-14-01053-t003:** Improved bread wheat traits through chromosomal introgression.

Member	Trait	Technique Used	Reference
Wheat landraces	Drought tolerance	Crossing and selection	[83]
Emmer wheat	Drought tolerance	Synthetic, backcross	[84]
*Agropyron elongatum*	Root development	In situ hybridization and backcrossed	[85]
Exotic germplasm (wheat landraces)	Root mass to deeper soil profiles	Interspecific hybridization	[86]
*Aegilops geniculata*	-	Interspecific hybridization	[86]
Wild emmer	Morphophysiological traits	Crossed with wild emmer (G18-16) and durum (Langdon)	[87]
*Triticum dicoccoides*	Drought tolerance	QTL analysis and positional cloning of QTLs	[88]
*Elymus semicostatus (Nees ex Steud.)*	Leaf sheath compactness, number of florets, spike curvature, spike density	Screening for morpho-physiological traits for drought tolerance	[89]
*Ae. Tauschii* (DD genome)	Cellular thermotolerance	Diploid Xhexaploid cross approach	[90]
*Aegilops tauschii*, *Triticum dicoccoides*	Root and shoot growth, membrane injury	Crossing and selection	[91]

**Table 4 plants-14-01053-t004:** Overview of QTLs/MQTLs associated with drought resistance in wheat.

Traits	MQTL	Chr (cM)	Position (cM)	Reference
Plant height	MQTL-PH1	1A	54.37	[136]
Root number	MQTL6	3A	45.8	[137]
Root volume, root surface area, root length	MQTL7	3A	75.5	[137]
CID, Col, KN, SD,	MQTL2	1A	60	[138]
SG, WSC, WS, Yld				
Photo, WSC	MQTL3	1A	89	[138]
Chlorophyll content	Qchl.ksu-3B	3B	67.2	[139]
Days to maturity	QDm-7D	7D	2.7	[140]
Days to heading	MQTL-HD3	5B	92.66	[136]
Stem reserve mobilization	QSrm.ipk-5D	5D	19	[141]
Stem reserve mobilization	QSrm.ipk-2D	2D	142	[141]
Grain yield	qGYWD.3B.2	3B	97.6	[142]
Grain yield	Qyld.csdh.7AL	7A	155.9	[142]
Thousand grain weight	QTgw-7D-b	7D	12.5	[140]
Drought tolerance	MQTL-DT2	4B	41.52	[136]

CID = carbon isotope discrimination, Col = coleoptile vigor, KN = kernel number, Photo = photosynthesis, SG = stay-green, SD = spike density, WS = water status, WSC = water soluble carbohydrates, Yld = yield, MQTL = Meta-QTL, Chr = chromosome, cM = centimorgan.

**Table 7 plants-14-01053-t007:** List of some omics databases in wheat crops.

Database	Database Salient Feature	URL
CerealsDB	Genotyping information for over 6000 wheat accessions and describe new webtools for exploring and visualizing the data and also describe a new database of quantitative trait loci that links phenotypic traits to CerealsDB SNP markers and allelic scores for each of those markers	https://www.cerealsdb.uk.net/cerealgenomics/CerealsDB/indexNEW.php (accessed on 22 March 2024)
WheatGmap	Wheat gene mapping	https://www.wheatgmap.org (accessed on 20 March 2024)
PmiRExAt	A new online database resource that caters plant miRNA expression atlas.	http://pmirexat.nabi.res.in (accessed on 15 May 2024)
Triti-Map	Wheat gene and regulatory elements mapping	http://bioinfo.cemps.ac.cn/tritimap/ (accessed on 22 March 2024)
expVIP	Wheat transcriptome resources for expression analysis	http://www.wheat-expression.com/ (accessed on 20 March 2024)
WheatExp	Homologue-specific database of gene expression profiles for polyploid wheat.	https://wheat.pw.usda.gov/WheatExp/ (accessed on 20 March 2024)
Wheat Panache	Wheat genome-wide copy number variations (CNVs) visualization	http://www.appliedbioinformatics.com.au/wheat_panache (accessed on 20 March 2024)
WheatGenome	Genome viewer with BLAST search portal, wheat auto SNPdb, links to wheat genetic maps and a wheat genome Wiki to allow interaction between diverse wheat genome sequencing activities	http://wheatgenome.info (accessed on 20 March 2024)
wDBTF	Collates 3820 wheat sTFs sequences	http://wwwappli.nantes.inrae (accessed on 20 March 2024)
MASWheat	Marker-assisted selection database for wheat	https://maswheat.ucdavis.edu/ (accessed on 20 March 2024)
WISP	The Wheat Improvement Strategic Program	http://www.wheatisp.org/ (accessed on 20 March 2024)
OpenWildWheat	Sequencing resources of *Ae. tauschii* accessions	https://openwildwheat.org/ (accessed on 20 March 2024)
Wheat Omics	Multi-omics data analysis	http://wheatomics.sdau.edu.cn/ (accessed on 20 March 2024)
Wheat Atlas	Atlas of wheat germplasm and production statistics	http://wheatatlas.org (accessed on 20 March 2024)
Wheat IS	An International Wheat Information System, supporting the wheat research community	http://www.wheatis.org/ (accessed on 20 March 2024)
Grain genes	Datasets useful to researchers working on wheat, barley, rye, and oat	https://wheat.pw.usda.gov (accessed on 22 March 2024)

## Data Availability

The current study did not involve the generation of new sequencing data. Therefore, there are no datasets generated or analyzed during the current study.

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
