# Peer review of "Wheat Drought Tolerance: Unveiling a Synergistic Future with Conventional and Molecular Breeding Strategies"

_plants, 2025, doi:10.3390/plants14071053_

Round 1
Reviewer 1 Report
Comments and Suggestions for Authors
The review is important, but it contains a lot of theory and very few practical examples. Also, it doesn’t lead or open to future research tips/suggestions. The study offers a good review of literature but with few examples that demonstrate theory.
Some suggestions to improve.
Table 1. Please indicate in table 1 what increased and decreased and by which percentage and their correspondent references.
Figure 1. Authors are invited to re-do the figure to be accurate: aren’t avoidance, tolerance and escape different survival mechanisms of plants? However, in the schema figure 1 these come from the stress factor but indeed all are for the survival for the plant, arrows and boxes are misplaced. Please correct and authors are invited to re-design the schema to make sense.
The title of table 2 needs to be improved and to show indeed the information contained, as it contains more than is referred to in the title. Same for table 5.
Please described briefly each figure and what is the purpose of the figure in relation with the text.
Section 4.1.2 gives too much description of what is backcrossing. Instead give more examples of where this methodology was indeed used for drought improvement in wheat and its effective results.
The same applied to section 4.1.3., there are no proper recent examples, and makes the section with little value for readers.
Please describe figure 8 in the legend and how to be used by breeders.
Author Response
Comment 1: Table 1. Please indicate in table 1 what increased and decreased and by which percentage and their correspondent references.
Response 1: Thank for Pointing out the action has been taken, please follow the revised manuscript.
Comment 2: Figure 1. Authors are invited to re-do the figure to be accurate: aren’t avoidance, tolerance and escape different survival mechanisms of plants? However, in the schema figure 1 these come from the stress factor but indeed all are for the survival for the plant, arrows and boxes are misplaced. Please correct and authors are invited to re-design the schema to make sense.
Response 2: Thank you for your suggestions, the figure re draw please follow the revised manuscript.
Comment 3: The title of table 2 needs to be improved and to show indeed the information contained, as it contains more than is referred to in the title. Same for table 5.
Response 3: Thank you for your pointing out the action has been taken please follow the revised manuscript.
Comment 4: Please described briefly each figure and what is the purpose of the figure in relation with the text.
Response 4: Thank you for your suggestions the action has been taken please follow the revised manuscript.
Comment 5: Section 4.1.2 gives too much description of what is backcrossing. Instead give more examples of where this methodology was indeed used for drought improvement in wheat and its effective results.
Response 5: Thank you for your pointing out the action has been taken please follow the revised manuscript.
Comment 6: The same applied to section 4.1.3., there are no proper recent examples, and makes the section with little value for readers.
Response 6: Thank you for your pointing out the action has been taken please follow the revised manuscript.
Comment 7: Please describe figure 8 in the legend and how to be used by breeders.
Response 7: Thank you for your pointing out the action has been taken please follow the revised manuscript.
Reviewer 2 Report
Comments and Suggestions for Authors
Dear Authors,
Reviewer comments plants-3488579
The review manuscript entitled „Wheat drought tolerance: unveiling a synergistic future with conventional and molecular breeding strategies“ represents a very useful overview on modern breeding approaches aimed at an enhancement of wheat drought tolerance. The focus is given on molecular approaches including marker-assisted selection (MAS), genome-wide association studies (GWAS), mutational studies and TILLING as well as genome-editing technologies (zinc-finger nucleases ZFN, TALEN, CRISPR/Cas). I think that the present manuscript provides a comprehensive overview on experimental approaches used for breeding new wheat varieties with improved drought tolerance.
I can recommend the manuscript for publication in Plants. I have only some minor suggestions on the present manuscript which are provided below:
1/ I think that in Introduction or somewhere at the beginning, major contrasting drought-response strategies, namely conservative, water-saving strategy and water-spending strategy, and possibilities of their utilization for various drought patterns in the environments should be discussed. I can recommend to study and cite the works of Tardieu et al., namely Tardieu (2012) J. Exp. Bot. 63: 25-31 or a review by Kosová et al. (2014) Czech J. Genet. Plant Breed. 50: 247-261.
2/ Regarding the omics approaches, I can suggest the authors to add some basic information on wheat drought proteomics studies since several proteins can be considered candidates associated with enhanced drought tolerance. I can suggest the authors to look at some studies on wheat drought proteomics with a focus either on a combination of drought and aphid infestation, e.g., Kosová et al. (2022) Front. Plant Sci. 13: 1005755, or a combined physiological, proteomic and metabolomic analysis on a long-term drought treatment followed by a recovery by Nešporová et al. (2024) Plant Stress 13: 100509.
3/ Formal comments on the text related to terminology, English language and style:
Abstract, line 12: Correct the typing error in the word „emphasizing“ (not „emphasizeing“).
Introduction, line 46: Add a comma following the words: „In the 2050,…“
Line 106: Correct the typing error in the word „authors“ (not „authores“).
Terminology:
Line 205: Correct the plant scientific name „Aegilops“ (not „Aegilopsis“).
Table 3: Correct the plant scientific name „Triticum dicoccoides“ (not „Triticeae dicoccoides“).
Line 294: Correct the term „cyclic selection“ (not „cyclical selection“).
Line 324: I think that the term should be „Quantitative trait loci (QTL) analysis“ NOT: „Qualitative trait loci (QTL) analysis“.
Line 337: Correct the typing error in the verb „validate“ (not „validat“).
Part 4.3.4. Genome editing and sequencing. Terminology: I think that the terminology should be corrected to „exone sequencing“ (NOT „exome sequencing“) – lines 561, 566.
In Table 7, the abbreviation „CNV“ should be explained in the table legend.
Line 630: Correct the verb form in the statement „Biomercator V3.0 based on Goffinet and Gerber´s algorithm are being used to conduct meta-analysis….“
Lines 638, 639: Add a space betwen the words „of“ and „developing“ and between the words „wheat“ and „in“ in the statement: „Combined with other novel technologies, this approach addresses crucial limitations of developing sustainable and climate-smart wheat in shortest timeframe……“
Conclusion, line 656: Correct the typing error in the word „wheat“ (not „whrat“).
Conclusion, line 662: Correct the typing error in the verb „generate“ (not „genesare“).
Final recommendation: Accept after a minor revision.

Author Response
Comment 1: 1/ I think that in Introduction or somewhere at the beginning, major contrasting drought-response strategies, namely conservative, water-saving strategy and water-spending strategy, and possibilities of their utilization for various drought patterns in the environments should be discussed. I can recommend to study and cite the works of Tardieu et al., namely Tardieu (2012) J. Exp. Bot. 63: 25-31 or a review by Kosová et al. (2014) Czech J. Genet. Plant Breed. 50: 247-261.
Response 1: Thank you for your suggestion; however, the manuscript already contains a sufficient number of references, so adding more may not be necessary.
Response 2: Regarding the omics approaches, I can suggest the authors to add some basic information on wheat drought proteomics studies since several proteins can be considered candidates associated with enhanced drought tolerance. I can suggest the authors to look at some studies on wheat drought proteomics with a focus either on a combination of drought and aphid infestation, e.g., Kosová et al. (2022) Front. Plant Sci. 13: 1005755, or a combined physiological, proteomic and metabolomic analysis on a long-term drought treatment followed by a recovery by Nešporová et al. (2024) Plant Stress 13: 100509.
Response 2: Thank you for your suggestion; however, the manuscript already contains a sufficient number of references, so adding more may not be necessary.
Comment 3: Formal comments on the text related to terminology, English language and style.
Response 3: Thank you for pointing out the action has been taken please follow the revised manuscript.
Comments 4: Abstract, line 12: Correct the typing error in the word „emphasizing“ (not „emphasizeing“).
Response 4: Thank you for pointing out the action has been taken please follow the revised manuscript.
Comments 5: Introduction, line 46: Add a comma following the words: „In the 2050,…“
Response 5: Thank you for pointing out the action has been taken please follow the revised manuscript.
Comment 6: Line 106: Correct the typing error in the word „authors“ (not „authores“).
Response 6: Thank you for pointing out the action has been taken please follow the revised manuscript.
Comment 7: Line 205: Correct the plant scientific name „Aegilops“ (not „Aegilopsis“).
Response 7: Thank you for pointing out the action has been taken please follow the revised manuscript.
Comment 8: Table 3: Correct the plant scientific name „Triticum dicoccoides“ (not „Triticeae dicoccoides“).
Response 8: Thank you for pointing out the action has been taken please follow the revised manuscript.
Comments 9: Line 294: Correct the term „cyclic selection“ (not „cyclical selection“).
Response 9: Thank you for pointing out the action has been taken please follow the revised manuscript.
Comment 10: Line 324: I think that the term should be „Quantitative trait loci (QTL) analysis“ NOT: „Qualitative trait loci (QTL) analysis“.
Response 10: Thank you for pointing out the action has been taken please follow the revised manuscript.
Comment 11: Line 337: Correct the typing error in the verb „validate“ (not „validat“).
Response 11: Thank you for pointing out the action has been taken please follow the revised manuscript.
Comment 12: Part 4.3.4. Genome editing and sequencing. Terminology: I think that the terminology should be corrected to „exone sequencing“ (NOT „exome sequencing“) – lines 561, 566.
Response 12: Thank you for pointing out the action has been taken please follow the revised manuscript.
Comment 13: In Table 7, the abbreviation „CNV“ should be explained in the table legend.
Response 13: Thank you for pointing out the action has been taken please follow the revised manuscript.
Comment 14: Line 630: Correct the verb form in the statement „Biomercator V3.0 based on Goffinet and Gerber´s algorithm are being used to conduct meta-analysis….“
Response 14: Thank you for pointing out the action has been taken please follow the revised manuscript.
Comment 15: Lines 638, 639: Add a space betwen the words „of“ and „developing“ and between the words „wheat“ and „in“ in the statement: „Combined with other novel technologies, this approach addresses crucial limitations of developing sustainable and climate-smart wheat in shortest timeframe……“
Response 15: Thank you for pointing out the action has been taken please follow the revised manuscript.
Comment 16: Conclusion, line 656: Correct the typing error in the word „wheat“ (not „whrat“).
Response 16: Thank you for pointing out the action has been taken please follow the revised manuscript.
Comment 17: Conclusion, line 662: Correct the typing error in the verb „generate“ (not „genesare“).
Response 17: Thank you for pointing out the action has been taken please follow the revised manuscript.
Round 2
Reviewer 1 Report
Comments and Suggestions for Authors
Please see comments attached

Author Response
Comment 1: The authors only changed the symbols, not doing anything of what was requested. It was
requested what changes in % exist, so the manuscript differentiates from others. Plus, where
are the references? Is the table a hypothetic table? If there is a copy from reference 388
please mention as source: 388.
Response 1: We appreciate the reviewer’s insightful suggestion. The traits listed in Table 1 are influenced by drought and indirectly affect seed yield. As most researchers focus on measuring percentage changes primarily for direct traits or seed yield, the specific data requested is not readily available in the literature. Therefore, we have compiled only the observed increases and decreases. However, as the reviewer has pointed out, we have now included the relevant data sources in the revised manuscript. We sincerely appreciate your thoughtful review and valuable guidance.
Comment 2: This is not seen in the manuscript, below each figure is exactly the same text.
Response 2: Thank you for your suggestions the action has been taken please follow the
revised manuscript.
Comment 3: Give the reference for the statement that was added: There are several examples like the
Njoro BW1 wheat varieties developed in Kenya, this variety emerged from gamma radiation-induced
mutations in the wheat genotype 'Kwale'. Njoro BW1 exhibits enhanced drought tolerance
and is recommended for cultivation in Kenya's marginal areas.
Response 3: Thank you for your suggestions the action has been taken please follow the
revised manuscript.
Comment 4: It looks very awkward the authors start a paragraph as they add in point 3.3. The same for
point 4.1.4. and the same for where they placed the Figure 5 description. All added text
were placed in portions of the text not making the connection.
Response 4: Thank you for your suggestions the action has been taken please follow the
revised manuscript.